# Regulation of miR319b-Targeted *SlTCP10* during the Tomato Response to Low-Potassium Stress

**DOI:** 10.3390/ijms24087058

**Published:** 2023-04-11

**Authors:** Xin Liu, Lingling Pei, Lingling Zhang, Xueying Zhang, Jing Jiang

**Affiliations:** 1College of Horticulture, Shenyang Agricultural University, Shenyang 110866, China; 2Key Laboratory of Protected Horticulture of Education Ministry, Shenyang 110866, China

**Keywords:** miR319b, root growth, low-potassium stress, *SlTCP10*, *SIJA2*, tomato plant

## Abstract

Potassium deficiency confines root growth and decreases root-to-shoot ratio, thereby limiting root K^+^ acquisition. This study aimed to identify the regulation network of microRNA319 involved in low-K^+^ stress tolerance in tomato (*Solanum lycopersicum*). SlmiR319b-OE roots demonstrated a smaller root system, a lower number of root hairs and lower K^+^ content under low-K^+^ stress. We identified *SlTCP10* as the target of miR319b using a modified RLM-RACE procedure from some SlTCPs’ predictive complementarity to miR319b. Then, *SlTCP10*-regulated *SlJA2* (an NAC transcription factor) influenced the response to low-K^+^ stress. CR-*SlJA2* (CRISPR-Cas9-*SlJA2*) lines showed the same root phenotype to SlmiR319-OE compared with WT lines. OE-*SlJA2*(Overexpression-*SlJA2*) lines showed higher root biomass, root hair number and K^+^ concentration in the roots under low-K^+^ conditions. Furthermore, *SlJA2* has been reported to promote abscisic acid (ABA) biosynthesis. Therefore, *SlJA2* increases low-K^+^ tolerance via ABA. In conclusion, enlarging root growth and K^+^ absorption by the expression of SlmiR319b-regulated *SlTCP10*, mediating *SlJA2* in roots, could provide a new regulation mechanism for increasing K^+^ acquisition efficiency under low-K^+^ stress.

## 1. Introduction

Potassium (K^+^) is a necessary macronutrient for normal plant growth and development. Furthermore, K^+^ plays an important role in a lot of physiological courses, such as photosynthesis, enzyme activation, membrane potential maintenance, osmoregulation and ion homeostasis [1]. The high and relatively stable content of K^+^ is the guarantee of the cellular compartments and its flow between different cells or tissues. K^+^ is supplied from soil to plants through root uptake. The roots of plants are in direct contact with the soil. Therefore, the root cells, especially root epidermal cells and root hair cells, first receive the K^+^ deficiency signal [2]. Plants react to the low-K^+^ levels by influencing root growth and structure in different ways, such as suppressing primary roots’ growth and promoting root hair elongation [3]. In plants, low-K^+^ signal transduction causes a downstream response and ultimately promotes their adaptation to K^+^ deficiency [4]. K^+^ deficiency can activate K^+^ uptake through the plant root by influencing the K^+^ transporters’ activity [5].

MicroRNAs (miRNAs) are a class of small RNA (21–24nt long) that regulate target expression by degrading mRNA [6]. In plants, after RNase III Dicer-like 1 cutting, the miRNA strand of the miRNA: miRNA* duplex is loaded into an AGO protein to catalyze mRNA cleavage or translational repression [7]. Most miRNAs are related to abiotic stress [8]. In the past few years, miRNAs have been found to be involved in plant responses to nutrient stresses [9]. However, few miRNAs have been investigated in the role response to low-K^+^ stress [10]. In *O. sativa*, miR399 is induced by low-K^+^ stress [11]. Compared to other varieties, the LA2711 tomato has a higher K^+^ uptake. The ABA content in its leaves is high and increases sharply under salt stress [12]. Moreover, miRNAs take part in the plant responses to nutrient deficiencies [11]. Compared to nitrogen, phosphorus, sulfur and copper stresses, studies on K^+^ stress are lacking [13]. miR399 expression levels could be induced by low-K^+^ levels and degradation of its target gene, LTN1/OsPHO2, leading to an increase in the K^+^ transport protein OsHAK25 [11]. Among the 20 conserved miRNA families in plants, miRNA319 is involved in plant leaf and flower organ growth and the regulation of hormone signaling pathways. Members of the miR319 family are highly conserved in different plants; for example, in *Arabidopsis thaliana*, miR319a and miR319b have identical sequences, while miR319c is different from miR319a and miR319b at the 3′end nucleotide [14]. The sequence of miR319b in tomato is the same as that of miR319a in *A. thaliana* [15]. miRNAs function by complementation and inhibition of transcription or translation. miR319 has been validated using 5′RACE to target transcription factors, such as TCPs and MYBs [14,16]. In *Arabidopsis*, miR319a primarily acts on TCPs, including TCP2, TCP3, TCP4, TCP10 and TCP24 [17]. In tomato, 24 TCP family transcription factors have been cloned [18]. TCPs comprise one plant-unique group of transcription factors. TCP proteins are named after three such domains found in different plants: teosinte branched1 of maize, CYCLOIDEA of Antirrhinum and PCNA promoter-binding factors [19,20]. TCPs can bind to promoters that contain the motifs GGACCA or TGGTCC [21]. TCP4, regulated by miR319, is available for cell proliferation and is a direct regulator of miR396 [17].

NAC family transcription factors play important roles in development and both abiotic and biotic stress responses [21]. AtNAC2 was identified to promote lateral root development, which can also be induced by ABA in response to salt stress [22]. GmNAC20 can enhance salt tolerance and promote later root formation in *Arabidopsis* [23]. Overexpression of GmNAC11 can improve salt tolerance [24]. Moreover, NAC transcription factors were differentially expressed during low-K^+^ stress in sugarcane [23]. NAC proteins are plant-specific transcription factors that are typically abiotic stress-responsive. OsNACs are induced under abiotic stress conditions in rice by jasmonic acid (JA) and partly by ABA [25]. JA2, an NAC transcription factor in tomato, activates ABA biosynthesis-related gene transcription, influencing ABA biosynthesis [26].

K^+^ is primarily absorbed by the plant roots. We previously reported that the low-K^+^-tolerant JZ34 and low-K^+^-sensitive JZ18 tomato genotypes display different root configurations under K^+^ deficiency stress [27]. Therefore, it is necessary to study the molecular regulation mechanism of the tolerance of JZ34 to low-K^+^ stress. This tolerance to low-K^+^ stress might be due to changes in the root growth signaling pathway, which in turn influences K^+^ absorption under K^+^ deficiency conditions. Notably, *SlmiR319b* expression levels in JZ34 are significantly lower than those in JZ18 during the early stage of low-K^+^ treatment. The discovery of how the target gene *TCP10* mediates the downstream pathway regulated by *SlmiR319b* provides new insight into the function of this important regulatory network.

## 2. Results

### 2.1. Expression Analysis of miR319b Involved in Regulating Low-K^+^ Stress in Tomato Plants

JZ34 tomatoes (low-K^+^-tolerant) showed better root development and K^+^ absorption under K^+^ deficiency conditions than JZ18 tomatoes (low-K^+^-sensitive) [27]. According to our previous study, *SlmiR319b* is differentially expressed between JZ18 and JZ34 tomatoes under K^+^ deficiency stress. Here, we performed qRT-PCR and observed that, in JZ18 tomatoes, the expression levels of *SlmiR319b* were higher under low-K^+^ stress conditions than under normal conditions at 4 h and 12 h (Figure 1). Conversely, the expression levels of *SlmiR319b* in JZ34 tomatoes were lower under low-K^+^ stress conditions than under normal conditions at 4 h and 8 h. Thus, *SlmiR319b* expression responded differentially to low-K^+^ stress between JZ18 and JZ34 at the early stage of low-K^+^ treatment (Figure 1).

### 2.2. The Expression of Related Genes and Root Phenotype in OE-miR319 Plants

To investigate the contribution of SlmiR319b to the response to low-K^+^ stress, we created 14 lines of SlmiR319b overexpression. We introduced the OE-miR319 construct into the JZ34 tomato cultivation. For JZ34, we further improved the genetic transformation method of tomato. Finally, after PCR screening and measurement of SlmiR319b expression levels, eight lines (L6, L7, L8, L9, L12, L14, L16, L17) showed a significant increase in SlmiR319b expression (Figure 2a). We also examined the target *SlTCP10* expression levels among the eight lines (Figure 2b). In the L6, L8, L9, L12 and L14 OE-miR319 plants, target *SlTCP10* showed the opposite tendency to *SlmiR319* expression levels. Therefore, these five lines were selected to save the next generation of seeds.

OE-miR319 had lower root length, volume and surface area than JZ34 under normal condition. Under low-K^+^ treatment, the root length, volume and surface were all decreased in JZ34 and OE-miR319, with OE-miR319 showing the most significant reduction percentage in root length (Table 1). Therefore, miR319b overexpression directly inhibited tomato root growth and the root length was more sensitive in OE-miR319 under low-K^+^ stress. The root system of JZ34 and OE-miR319 plants gradually developed and became larger with the normal K^+^ concentrations (4 mM) treatment for 0, 3 and 7 d (Figure 2c). However, after 7 d of treatment at low-K^+^ concentrations (0.5 mM), the root system of SlmiR319b-OE plants appeared significantly smaller than that of JZ34. Micro-examination revealed that SlmiR319b-OE plants had less root hair than JZ34, especially after 7 d of low-K^+^ treatment (Figure 2d). The number of root hairs was counted under the same microscope field of vision. The number of root hairs was significantly lower in OE-miR319 than JZ34 under low-K^+^ stress (Table 2). The number of root hairs was sharply increased in JZ34 under low-K^+^ treatment for 3 and 7 d. Under low-K^+^ treatment for 7 d, the root–shoot ratios of OE-miR319 plants were found to be significantly lower than those of JZ34 (Figure 2e). Under normal conditions, the root shoot ratios of OE-miR319 did not change much with the growth for 3 and 7 d. Further analysis of the K^+^ content in roots (Figure 2f) revealed that OE-miR319 plants exhibited significantly lower K^+^ content than JZ34 under K^+^ deficiency stress for 3 or 7 d. The K^+^ content of OE-miR319 was also lower than that of JZ34 under the normal condition. The root K^+^ content of JZ34 under low-K^+^ stress was nearly unchanged or even increased compared with that under normal condition. Thus, JZ34 had the strong root adaptability to improve the K^+^ absorption under the low-K^+^ stress, but SlmiR319b overexpression actually inhibited the root growth prior to low-K^+^ stress, decreasing initial K^+^ absorption and leading to low-K^+^ content not only under low-K^+^ stress but also under normal conditions.

### 2.3. SlTCP10 Was Targeted by SlmiR319b

SlTCP2 (*Lanceolate*) is cleaved at nucleotides 10 and 11 of the miR319 recognition site [27]. According to our results, the transcription factors Solyc07g053410.2, Solyc02g077250.2 and Solyc12g014140.1 (*SlTCP10*, *SlTCP1* and *SlTCP3*, respectively) (Tomato Sol Genomic Network database, http://solgenomics.net/, accessed on 3 April 2023) were predicted to serve as targets for SlmiR319b using complementary sequence analysis (Figure 3a). To confirm that these three TCPs were subjected to miRNA-mediated cleavage in vivo, we isolated mRNAs from tomato leaves and performed 5′-RACE to detect their 5′ cleavage products. A single 5′-RACE product of the size predicted to be generated from the cleaved *SlTCP10* template could be amplified, and the products from 5′RNA ligase-mediated RACE corresponded to the miRNA cleavage between nucleotides 10 and 11 of the miR319 recognition site in *SlTCP10* transcripts (Figure 3b–d). However, the 5′-RACE products of the size predicted to be generated from the cleaved *SlTCP1* and *SlTCP3* template could not be amplified.

### 2.4. SlmiR319b Targets SlTCP10, Mediating SlJA2 to Influence K^+^ Absorption

To identify the signaling components downstream from the *SlTCP10* response to low-K^+^ stress in tomatoes, we analyzed the RNA-Seq data of JZ18 and JZ34 treated with low-K^+^ stress, which led us to hypothesize that a related NAC family transcription factor, *SlJA2* (jasmonic acid 2), may be involved. *SlJA2* expression was induced at the low-K^+^ treatment times of 4, 8, 12 and 24 h in the low-K^+^-tolerant JZ34 (Figure 4a), indicating that *SlJA2* may be an important factor in enhancing this tomato’s low-K^+^ tolerance. The *SlJA2* promoter was found to contain the TCP motif TGGTCC at position −254 bp.

The interactions between TCP10 and *SlJA2* were investigated further. We performed EMSAs with an in vitro-translated TCP10 protein. The 100 bp fragment containing the TGGTCC element included in the JA2 promoter sequence was labeled with biotin. We first constructed a *SlTCP10* and pET28a fusion recombinant plasmid. After sequencing, the plasmid was transformed into the BL21 bacterial strain for expression. A *SlTCP10* fusion His recombinant protein was obtained by induction and purification. The EMSA indicated that the His-TCP10 protein bound the oligo DNA with molecules containing the *SlJA2* promoter fragment (Figure 4b), which suggests that *SlTCP10* could bind the promoter with TGGTCC to regulate the expression of *SlJA2*. A Y1H assay was performed to test the association between SlTCP10 and the SlJA2 promoter. The results showed that *SlTCP10* could associate with the ‘TGGTCC’ element located at −1798 bp and −1803 bp of the *SlJA2* promoter (Figure 4c) and the mutation in these binding sites could break the association, supporting the hypothesis that *SlTCP10* is a transcription factor associated with *SlJA2*.

### 2.5. SlJA2 Influenced the Tomato Low-K^+^ Tolerance via ABA

JA2 activates the ABA biosynthetic gene NCED1, which encodes 9-cis-epoxycarotenoid dioxygenase, a rate-limiting enzyme of ABA biosynthesis [28,29]. Therefore, we suspected that *SlJA2* could affect tomato response to low-K^+^ stress via ABA. 25 μmol/L ABA was added to the tomato Ailsa Craig ‘AC’ culture medium. This showed that the root system following treatment with ABA was larger than following the normal treatment under low-potassium stress (Figure 5a). However, this ABA-promoting effect on the root did not occur in the absence of low-K^+^ stress under the CK condition. The concentration of K^+^ in root and shoot were both increased with ABA treatment under low-K^+^ stress (Figure 5b). The classical ABA-deficient *sit* mutant of tomato harbors a mutation in an AO enzyme that catalyzes the final step of ABA biosynthesis [28,30]. We treated *sit* mutants with low-K^+^ treatment. After 7 d low-K^+^ treatment, the whole plants and roots of the *sit* mutants appeared weaker than the WT plants (Figure 5c,d). After 21 d low-K^+^ treatment, the *sit* mutants had more yellow leaves with particularly yellow edges (Figure 5e). The mutants seemed to be suffering more severe K^+^ stress than the WT plants. Considering the key role of *SlJA2* in ABA synthesis, this suggests that *SlJA2* can affect the response of tomatoes to low-K^+^ stress via ABA.

### 2.6. The Key Regulator SlJA2 Influenced the Tomato Low-K^+^ Tolerance

SlJA2 appears to be the major factor regulated by TCP10. Therefore, *SlJA2* plays a key role in the miR319 regulatory network of response to low-K^+^ stress in tomato plants. We further evaluated the role of *SlJA2* in response to low-K^+^ stress by knocking down and overexpressing it. We used CRISPR-Cas9 to generate transgenic tomato lines (CR-*SlJA2*) (Figure 6a). We observed four single-base mutations, one double-base mutation, one single-base insertion and one three-base deletion in CR-*SlJA2*-20 by sequencing the positive plants (Figure 6b). We found one single-base mutation, one single-base insertion and a six-base deletion in CR-*SlJA2*-5 by sequencing the positive plants (Figure 6b). Simultaneously, the pBWA(V)HS-*SlJA2* overexpression vector was constructed. Expression levels in the overexpressing plants were detected using SlJA2 RT-PCR (Figure 6c). We observed that the root development of the OE-*SlJA2* and CR-*SlJA2* lines exhibited greatly contrasting states under control conditions (LK-0d) (Figure 6d). The OE-*SlJA2* plants had a larger root system than the CR-SlJA2 plants. After 7 d of low-K^+^ treatment, the OE-*SlJA2* lines still had more roots compared with the control; conversely, the CR-*SlJA2* lines showed fewer roots than the control. The OE-*SlJA2* plants had longer and more root hair after low-K^+^ treatment at 7 d than at 0 d (Figure 6e). By contrast, the CR-*SlJA2* plants had nearly no root hair after 7 d low-K^+^ treatment. Statistical analysis revealed that OE-*SlJA2* had significantly more root hairs than CR-*SlJA2* (Table 3). After low-K^+^ treatment for 7 d, the number of root hairs in CR-*SlJA2* and OE-*SlJA2* both declined, with CR-*SlJA2* showing a greater decline rate. The root length, volume and surface area data revealed that low-K^+^ stress expanded the root system of the OE-*SlJA2* lines but contracted that of the CR-*SlJA2* lines (Table 4). After 7 d low-K^+^ treatment, yellow leaves quickly appeared in the CR-*SlJA2* lines, but the OE-*SlJA2* and WT (AC) lines did not exhibit this typical K^+^ deficiency phenotype (Figure 6f). The root K^+^ contents of CR-*SlJA2* and AC were all decreased after 7 d low-K^+^ treatment compared with the normal condition (Figure 6g). However, the degree of reduction of the CR-*SlJA2* plants was greater. The root K^+^ content of OE-*SlJA2* with low-K^+^ treatment was nearly the same as in the control. In general, *SlJA2* enhanced tomato plant resistance to low-K^+^ stress and root system adaptation which was evidenced by the root growth and root hair development.

## 3. Discussion

The strong ability of the roots to take up K^+^ from the soil is a prerequisite for plant survival under low-K^+^ conditions [31,32]. The OE-miR319 plants showed slightly weaker roots than JZ34, especially under low-K^+^ treatment (Figure 3c). The OE-SlmiR319b plants had significantly shorter root length, volume and surface area than JZ34 (Table 1). The OE-SlmiR319b plant leaves exhibited a typical K^+^ deficiency phenotype, with the leaf margins turning yellow under low-K^+^ treatment at 21 d, unlike the leaves of JZ34 (Appendix A). Thus, JZ34 had stronger root adaptability than the OE-SlmiR319b plants to low-K^+^ stress. The CR-*SlJA2* plants had a weaker root system than the WT plants and the OE-*SlJA2* plants had a stronger root system than the WT plants (Figure 6d). The root phenotype data showed that the OE-*SlJA2* plants were more tolerant to K^+^ deficiency than CR-*SlJA2* plants (Table 4). K^+^ is primarily absorbed by the plant roots. K^+^ deficiency in plants confines root growth and decreases the root-to-shoot ratio, thus limiting root K^+^ acquisition [33]. K^+^ deficiency can affect both shoot and root growth [31,34]. In the roots, it impairs lateral root initiation and development [35]. It also appears to have a depressive effect on primary root growth [31,36,37]. Enlarging the root system and the development of adventitious roots could improve K^+^ uptake efficiency at low-K^+^ supply levels [33]. Our study was the first to discover that *SlmiR319b,* by targeting *SlTCP10,* plays an important role in maintaining tomato root growth under low-K^+^ stress. *SlJA2*, downstream from *SlTCP10*, plays the opposite role to *SlmiR319b*. *SlmiR319b*, by targeting *SlTCP10* and *SlJA2*, affected root and root hair growth as well as root K^+^ content (Figure 2f and Figure 6g). Therefore, preventing the root growth retardation caused by low-K^+^ levels could enhance the adaptation of tomato plants to K^+^ deficiency.

There is much research about miR319b-targeted TCPs’ involvement in the abiotic stresses. In rice, OsPCF6 and OsTCP21 showed obvious cold-induced expression, and overexpression, of miR319b, which markedly inhibited this induction [38]. Overexpression of OSa-miR319a improved drought and high-salinity tolerance [39]. It was also found that miR319 and its target TCPs were responsive to drought and salt stress in *Brassica napus* [40]. We added the function of miR319b-SlTCP10 response to low potassium for filling the gap of miR319-regulated TCPs mode in plant abiotic stress. The core of our response to low-K^+^ stress by miR319b-TCP10 was *SlJA2* transcription factor. The JA2 promoter contains no G-box-like motif (CACGTA), which is required for JA/COR induction [41]. However, sequence analyses have revealed the existence of two ABA-responsive element-like motifs (ACGTGTC) in the 2000-bp region of the JA2 promoter [26]. This indicates that the low-K^+^ response pathway mediated by SlJA2 is part of the ABA pathway. SlJA2 regulates the transcription of NCED1, which encodes a rate-limiting enzyme in ABA biosynthesis in tomatoes [26]. Using the classical ABA-deficient *sit* mutant of tomato [42], it was found that JA2 and NCED1 expression levels are markedly reduced in *sit* mutants compared with those in wild-type plants. Moreover, the *sit* mutants showed lower tolerance to low-K^+^ stress, with a smaller root system and less root hair (Appendix A). ABA, a stress-responsive hormone, has been widely studied for its effect on K^+^ transport. In isolated barley roots, ABA application has been shown to reduce the outflow of K^+^ from the stele cells to the xylem [43]. ABA can induce the expression of the K^+^ release channel gene GORK in *Arabidopsis* in response to water stress [44]. In the root system, the activity of the K_out_ channel decreases under the influence of ABA, whereas the K_in_ channel is activated by ABA treatment [44]. ABA enhances the selective absorption of K^+^ in cucumber by acting on the ion transporters and proton pumps at the gene expression level [45]. The application of ABA on the wild-type tomatoes demonstrated that ABA can promote the absorption of K^+^ and root growth under low-K^+^ stress (Figure 5a,b). In addition, FvTCP9 was reported to be involved in the biosynthesis of ABA [46]. Therefore, under low-K^+^ stress, *SlJA2* regulates K^+^ transport and absorption by affecting ABA biosynthesis. Accordingly, improving the regulatory pathway of ABA signaling in response to low-K^+^ stress in tomatoes needs to be investigated further.

This study identified the involvement of the miR319b regulation network in response to low-K^+^ stress in tomato plants. Most research about miRNAs involved in low-K^+^ stress were identified by the expression profile of miRNAs [13]. miR444 and its targets (MADS transcription factors) expression was in response to K^+^ deprivation [47]. miR168-AGO1 could also influence the downstream miRNAs expression to regulate the response to low-K^+^ stress in tomato [7]. The potassium uptake mediated by miRNAs still remains to be investigated in detail in the future. Currently, we determined a new target of SlmiR319b—SlTCP10—apart from *SlTCP4* and *SlTCP2* (*Lanceolate*) [15,27]. The OE-SlmiR319b plants had less root hair and a smaller root system, especially under low-K^+^ stress. Moreover, *SlTCP10* importantly induced downstream *SlJA2*, a typical NAC transcription factor. SlJA2 was known to increase the ABA content. The OE-SlmiR319b and CR-*SlJA2* lines showed lower tolerance to low-K^+^ stress than their wild-type counterpart. Therefore, SlmiR319b targeted *SlTCP10*, which in turn mediated SlJA2, thereby influencing the tomato response to low-K^+^ stress via the ABA pathway (Figure 7). As well, according to the miR168-AGO1 mode, miR319b targeting of *SlTCP10* may be the downstream miR168-AGO1 mode regulating the response to low-K^+^ response.

## 4. Materials and Methods

### 4.1. Plant Materials and Growth Conditions

Two tomato genotypes, ‘JZ34′ (low-K^+^-tolerant) and‘JZ18′ (low-K^+^-sensitive) (which were obtained in our lab by higher generation inbred lines and introduced in detail about the low-K^+^ tolerance by Zhao et al. in 2018 [27]), *sit* mutants (deficient in ABA biosynthesis) and transgenic tomatoes with their wild-type AC were planted in a greenhouse with an environment of 26/18 °C (day/night) and a light cycle of 16/8 h (light/dark). The root system of 25 d seedlings was cleaned with clean water, washed 3 times with distilled water, then transferred to a 12 L nutrient solution tank. Every tank was full of nutrient solution containing 4 mM KNO_3_, 1.5 mM Ca(NO_3_)_2_·4H_2_O, 2 mM MgSO_4_·7H_2_O, 0.67 mM NH_4_H_2_PO_4_, 0.05 mM H_3_BO_3_, 0.7 mM ZnSO_4_·7H_2_O, 0.009 mM MnSO_4_·4H_2_O, 0.32 mM CuSO_4_·5H_2_O, 0.05 mM FeSO_4_·7H_2_O, 0.04 mM Na_2_-EDTA and 0.1 mM (NH_4_)2MoO_4_. The nutrient solution pH was adjusted to 5.8 ± 0.1. The nutrient solution was replaced every 3 days. The concentration of KNO_3_ in nutrient solution was reduced from 4 mM (normal control K^+^) to 0.5 mM (K^+^ deficiency treatment) and the K^+^ deficiency treatment test was conducted. 25 μmol/L ABA was separately added into the nutrient solution as part of the JA treatment and ABA treatment. After 7 days of the low-K^+^ treatment, different tissues were obtained to measure the plant length, K^+^ content and weight.

### 4.2. Quantitative Real Time (qRT)-PCR Assay

Total RNA of the samples was extracted by TRIzol (Takara, Dalian, China) and the RNA was treated with RNase-free treatment (Takara). RQ1 DNase I (Promega, Madison, WI, USA) was used to remove the genomic DNA of the RNA (4 μg). The reverse transcription and RT-PCR of miRNA and cDNA protocols were derived from Liu et al. [48]. All experiments were replicated three times and RNA samples were obtained from a mixture of three different biological samples. The primers used in this study are listed in Appendix A.

### 4.3. miRNA Cleavage-Site Mapping

The target gene of the miRNA cleavage site was determined by a modified RLM-RACE method, as described by Hendelman et al. [49]. A total of 300 μg mRNA was obtained from the TRIzol RNA. This mRNA was used to obtain the RLM-RACE cDNA. The cDNA was first amplified by the primer GeneRacer 5′ and *SlTCP10*-RACE. Subsequently, the nested PCR was performed using the primer pair GeneRacer 5′ nested and *SlTCP10*-RACE_nested.

### 4.4. Promoter Analysis

We used the 2 kb sequence, which was located at a position 5′ of the start codon, to obtain the upstream sequences for *SlJA2*. The Tomato Sol Genomic Network (http://solgenomics.net/, accessed on 3 April 2023) and EnsemblPlants (plants.Ensemble.org/index.html, accessed on 20 August 2019) databases were used to obtain the gene and promoter sequence, respectively. PlantCARE (http://bioinformatics.psb.ugent.be/webtools/plantcare/html/, accessed on 3 April 2023) was used to predict the promoter elements.

### 4.5. Yeast One-Hybrid (Y1H) Assays

According to the manufacturer’s proposal, the Matchmaker Gold Yeast One Hybrid System Kit (Clontech) was used for Y1H determination. The *SlTCP10* and *SlJA2* CDs were connected to pGADT7 to generate AD-TCP10 and AD-JA2 constructs. The *SlJA2* promoter fragments were separately connected to the pAbAi vector to produce the pAbAi-bait plasmid. The pAbAi-bait plasmids were linearized and then transformed into Y1HGold yeast strains. The transformed yeast strains were selected on the plate by a selective synthetic glucose medium lacking uracil. Then, AD-TCP10 and AD-JA2 plasmids were transformed into Y1H Gold strains containing pAbAi-bait by screening on the SD/- Ura/ureobasidin A (AbA) plate. All transformations and filters were repeated three times.

### 4.6. Electrophoretic Mobility Shift Assay (EMSA)

Probe preparation: The specific 50 bp short segment on the promoter DNA sequence was selected and the biotin labeling synthesis was carried out by Shanghai Invitrogen Biological Company (Appendix A). The primer solution was diluted to 100 μM. The chemiluminescence EMSA kit from Biotech was used. Electrophoresis: A tank of 0.5 × TBE was precooled at 4 °C in advance and the electrophoresis tank was placed in an ice box for electrophoresis. The electrophoresis voltage was 100 V, so that the sample solution reached 1/4 of the gel and the electrophoresis was stopped. Film transfer: A nylon film with a size similar to that of the adhesive was soaked in 0.5 × TBE for at least 10 min. The nylon film was combined with the adhesive and the upper and lower layers of the adhesive and nylon film were padded with filter paper to ensure that there were no bubbles between the four layers of material. The film transfer tank was placed in the ice box and the current set to 380 mA for 40 min. UV cross-linking: The nylon membrane and the filter at the bottom of the membrane were placed in a clean petri dish, the UV ultra-clean workbench was opened and the petri dish was placed 10 cm away from the UV lamp for 8 min. After rinsing the cross-linked nylon film, the development was performed using an imager.

### 4.7. Vector Constructs and Tomato Transformations

The SrmiR319b precursor and *SlJA2* were amplified by PCR using gene specific primers. The SrmiR319b precursor and the *SlJA2* sequence were cloned into pCAMBIA3301/Luc and pBWA (V) HS/Gus, respectively, which contained two 35S cauliflower mosaic virus promoters, with the kanamycin resistance marker genes Hygromycin resistance and luciferase (Luc). CRISPR-cas9 vector targeted two sites of the first exon of *SlJA2* ORF. The sgRNAs were designed on CRISPR-PLANT (http://crispr.hzau.edu.cn/cgi-bin/CRISPR2/CRISPR, accessed on 21 June 2020). Colony PCR, restriction enzyme digestion and sequencing were used to screen positive recombinant plasmids. The vector was then transformed into the Agrobacterium tumefaciens GV3101 strain by electroporation. The Ailsa Craig (AC) material of tomato was used for genetic transformation. The strain with SlmiR319-OE vector was used for tomato JZ34 (low-K^+^-tolerant tomato). The target gene expression, sequence and the presence of strip marker genes of the T1 transgenic materials obtained were evaluated using qRT-PCR and sequencing techniques. Appendix A lists all primers used.

### 4.8. Root Phenotypic Analysis

The roots of the plants were washed with distilled water to remove the impurities after nutrient solution culture. The roots were laid in the transparent water tank in the root scanner (WINRhizo REG 2009) and distilled water was added to ensure that the roots did not cross themselves. The roots were spread evenly using a brush to avoid root overlap. The WinRhizo reg 2009 root scanning analysis system was used with the root scanning option to save the pictures and then select them to analyze a certain area.

The cover glass and slides were sterilized with alcohol in advance. The roots of the tomatoes were treated with the nutrient solution and washed with distilled water to remove the impurities. The root hairs were randomly selected and made into temporary slides, which were observed using an ordinary optical microscope (Nikon Eclipse 80i). The cover glass and slides were sterilized with alcohol in advance. The roots of the tomatoes were treated with the nutrient solution and washed with distilled water to remove the impurities. The root hairs were randomly selected and made into temporary slides, which were observed using an ordinary optical microscope (Nikon Eclipse 80i).

### 4.9. Ion Content Determination

The aboveground parts and roots of the tomatoes were washed, dried at 100 °C for 1 h and heated to 65 °C for 48 h. After the samples were dried, the upper and underground parts were placed in a mortar and ground into powders separately. An amount of 0.05 g of these powders was weighed and put into 10 mL centrifuge tubes separately. Next, 2 mL of hydrochloric acid (0.5 mol/L) was put into the tubes which were soaked at 25 °C for 3 d and turned up and down 8–10 times every 12 h. After the extraction, 5 mL of ddH_2_O was put into every tube, mixed well and filtered using a funnel and qualitative filter paper. The filtered stock solution was diluted 10-fold and the K^+^ concentration was measured with a flame spectrophotometer.

## 5. Conclusions

SlmiR319b and its target, *SlTCP10*, mediate *SlJA2* expression, thereby integrating hormonal ABA signaling to regulate low-K^+^ tolerance in tomato plants. Under low-K^+^ stress, miR319 targets *SlTCP10*, which directly interacts with *SlJA2*. In addition, *SlJA2* also induces endogenous ABA biosynthesis and ABA, in turn, improves the root growth and K^+^ content under low-K^+^ stress conditions. This miRNA regulatory mechanism provides new insight into the molecular networks governing the tomato response to low-K^+^ stress.

## Figures and Tables

**Figure 1 ijms-24-07058-f001:**
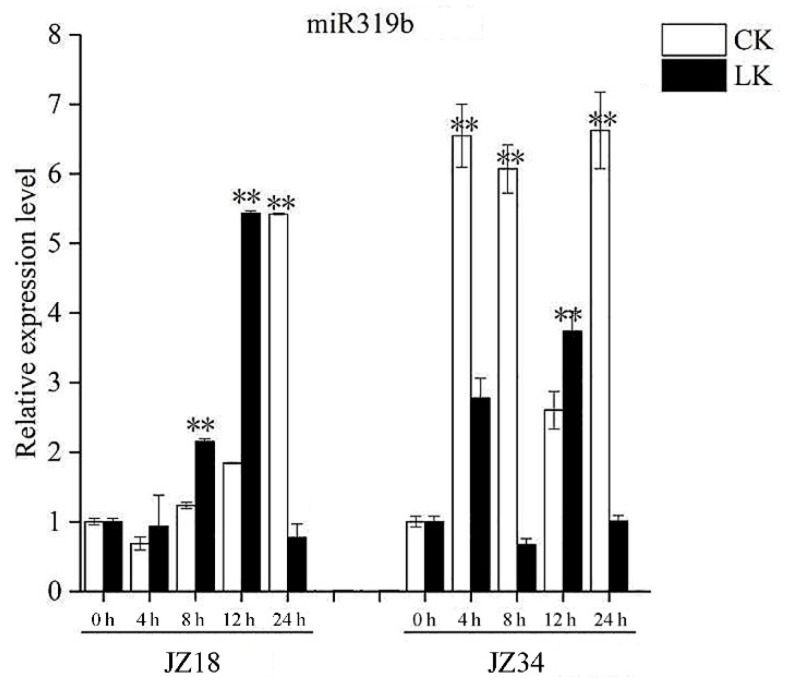
SlmiR319b and TCPs involved in response to low-potassium (K^+^) stress. Low-K^+^-sensitive JZ18 and low-K^+^-tolerant JZ34 seedlings were grown in the total nutrient solution ‘CK’ and subjected to low-K^+^ treatment with KNO_3_ (0.5 mM) ‘LK’. Quantitative real-time (qRT)-PCR was used to examine *SlmiR319b* expression under low-K^+^ stress at the treatment times of 0, 4, 8, 12 and 24 h. RNA was extracted from the roots. Three biological experiments from independent RNA extractions for each group of roots were analyzed. Values represent means ± SE. Asterisks indicate significant difference as determined using one-way ANOVA (**, *p* < 0.01).

**Figure 2 ijms-24-07058-f002:**
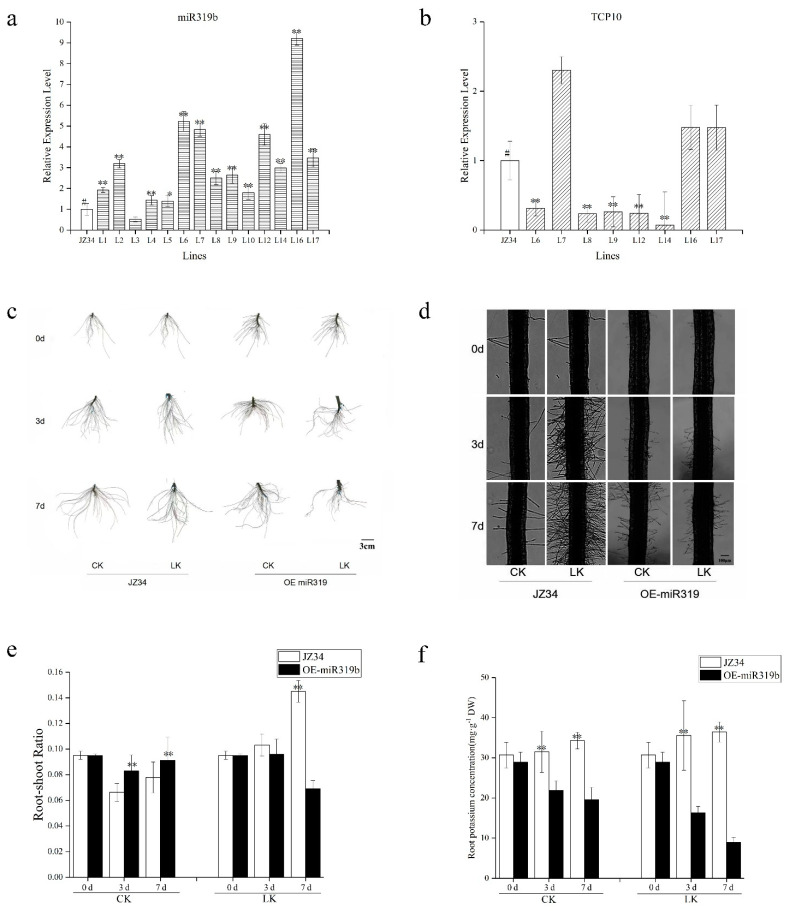
SlmiR319-OE T1 plants were obtained for the treatment of low-potassium (K^+^) stress. (**a**) The expression levels of SlmiR319 in the 13 SlmiR319-OE lines. Three biological experiments from independent RNA extractions for each group of leaves were analyzed. #, represents the control. (**b**) The expression levels of *SlTCP10* in the 8 SlmiR319-OE lines. Three biological experiments from independent RNA extractions for each group of leaves were analyzed. (**c**) Root phenotypes and (**d**) root hair of the SlmiR319b-OE lines compared with the WT(JZ34) lines supplied with a different concentration of K^+^ after 25 days of germination; the total nutrient solution is represented as ‘CK’ and the low-K^+^ treatment with KNO_3_ (0.5 mM) is represented as ‘LK’. The treatment time points were 0, 3 and 7 d. Bars = 3 cm/100 μm. (**e**) Fresh samples of the JZ34 and SlmiR319-OE aboveground (stems and leaves) and belowground parts were weighed at CK and LK treatment time points 0, 3 and 7 d. (**f**) The K^+^ content in the SlmiR319-OE and JZ34 roots determined at the CK and LK treatment time points 0, 3 and 7 d. Three biological replicates were prepared. Values represent means ± SE. Asterisks indicate significant difference as determined using one-way ANOVA (*, *p* < 0.05; **, *p* < 0.01).

**Figure 3 ijms-24-07058-f003:**
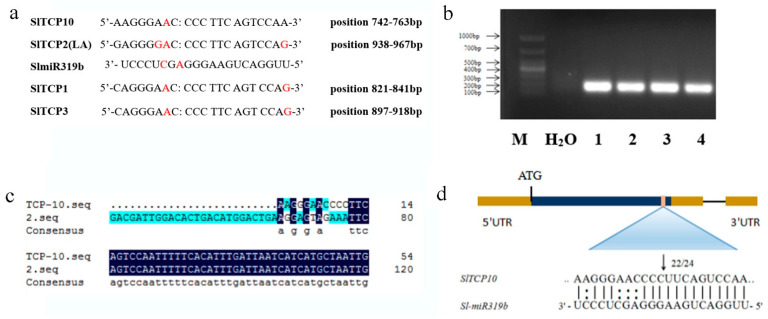
Cleavage analysis of *SlTCP10* transcript. The microRNA (miRNA)-mediated cleavage site was determined using RLM-RACE of total tomato leaf RNA using the RACE primer specific for the 3′ terminator. (**a**) Sequence of the miR319b recognition site in the *SlTCP2*, *SlTCP1* and *SlTCP10* mRNA. *SlTCP2* (*La*) has been reported to be the target of miR319. Red letters represent sequences that are not complementary to miR319 (**b**) The first amplification of the complementary DNA (cDNA) end (RLM-RACE) product (187 bp) was generated by using the GeneRacer 5′ primer for the 5′ end and *SlTCP10*-RACE primer for the 3′ end; the second amplification of the cDNA end product (145 bp) was generated by using the GeneRacer 5′ primer for the 5′ end and *SlTCP10*-RACE_nested primer for the 3′ end. (**c**) The sequencing analysis for the 5′-RLM-RACE product. (**d**) The single 5′-RACE product of the size predicted to be generated from cleaved *SlTCP10* template could be amplified and the sequence of 24 independent inserts revealed that 22 of their 5′ ends terminated at a position that paired with the 11th SlmiR319b nucleotide.

**Figure 4 ijms-24-07058-f004:**
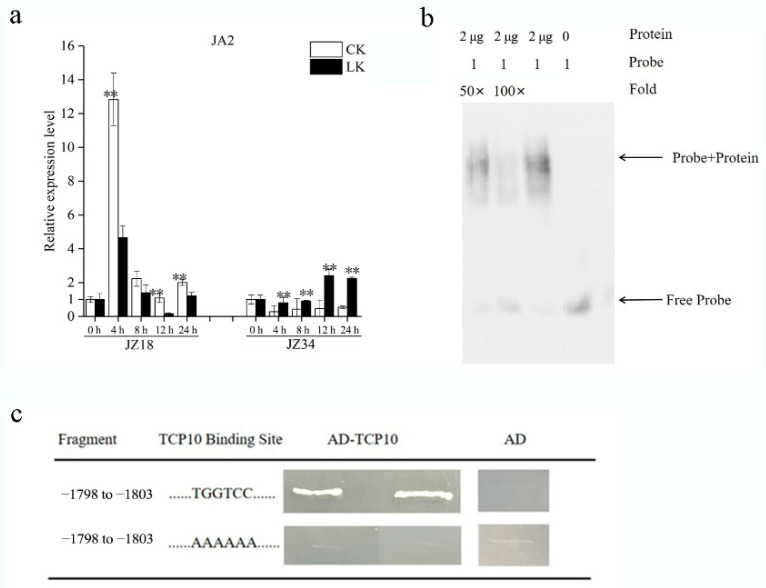
SlJA2 was regulated by *SlTCP10*, which is targeted by *SlTCP10*. (**a**) The expression of *SlJA2* in low-potassium (K^+^)-sensitive ‘JZ18′ and low-K^+^-tolerant ‘JZ34′ under low-K^+^ conditions at the treatment times of 0, 4, 8, 12 and 24 h. Three biological replicates were prepared. Values represent means ± SE. Asterisks indicate significant difference as determined using one-way ANOVA (**, *p* < 0.01). (**b**) EMSA of in vitro-translated SlTCP10 protein binding to the −225 to −275 fragment of the SlJA2 promoter (TGGTCC). From left to right, lane 1 and lane 2: 50× and 100× volumes cold probe competitive test groups. The protein in the cold probe competitive binding system without biotin labeling could weaken the binding band of biotin labeling probe and target protein; Lane 3: without competitive probes; Lane 4: control without SlTCP10 protein. (**c**) The association of SlTCP10 with the promoter of *SlJA2* using yeast one-hybrid (Y1H) assay. The *SlJA2* gene promoter element was mutated to ‘AAAAAA’ as a control and interacted with SlTCP10. Each fragment of the SlJA2 promoter was ligated to the pAbAi vector to generate pAbAi-bait plasmids, which were then linearized and transformed into the Y1HGold yeast strain and selected on a plate containing selective synthetic dextrose medium lacking uracil. AD-SlTCP10 was transformed into the Y1HGold strain holding pAbAi-bait and screened again on a plate with selective synthetic dextrose medium lacking uracil/AbA. The empty pGADT7 vector (AD) was transformed into a negative control.

**Figure 5 ijms-24-07058-f005:**
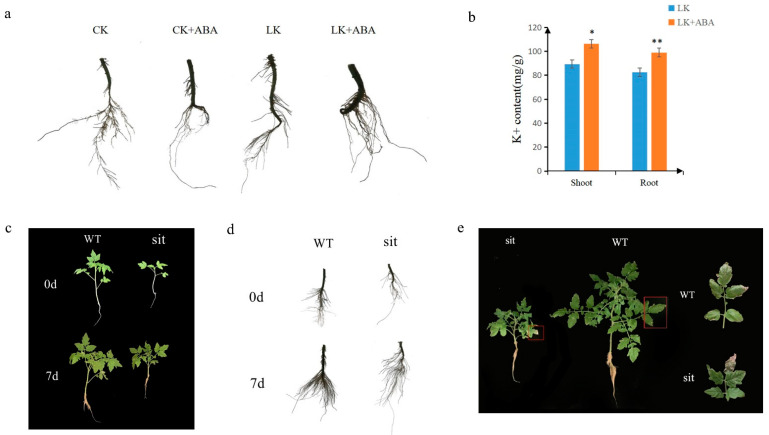
The influence on the tolerance of tomato to low potassium by the ABA treatment: (**a**) the root phenotype of under the normal condition (CK) and low potassium (LK) with ABA treatment 7 d; (**b**) the content of shoot and root under the normal condition (CK) and low potassium (LK) with ABA treatment for 7 d. The plant and root phenotype, root hair of *sit* mutants with low-potassium treatment for 7 days. WT represents the wild-type; (**c**,**d**) the *sit* mutants and WT with treatment of low potassium for 7 d; (**d**) the root phenotype of *sit* mutants and WT under the normal condition and low potassium (LK) treatment; (**e**) the plant phenotype of *sit* mutants and WT after low-potassium treatment for 21 d. Asterisks indicate significant difference as determined using one-way ANOVA (*, *p* < 0.05; **, *p* < 0.01).

**Figure 6 ijms-24-07058-f006:**
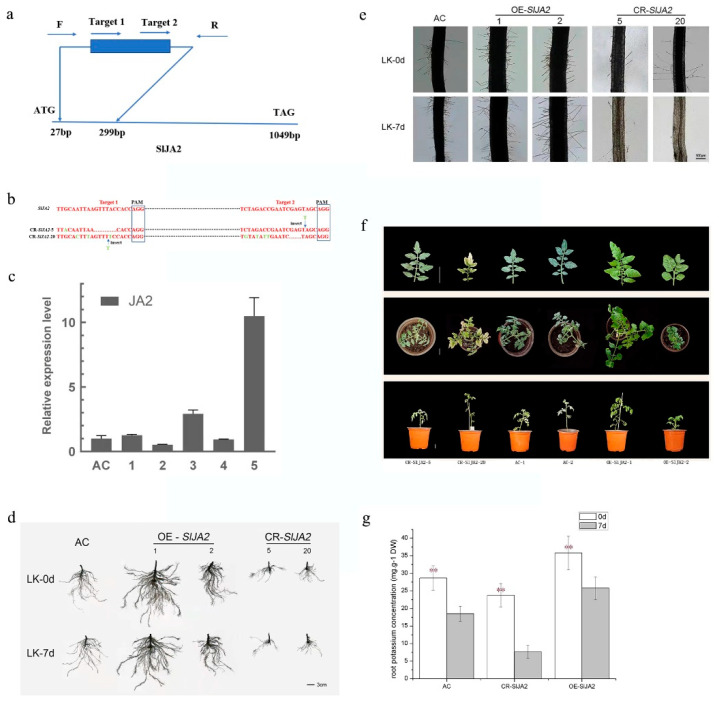
*SlJA2* affects the resistance of tomato plants to low-potassium (K^+^) stress. (**a**) Schematic illustration of the two sgRNA target sites in *SlJA2*. Blue arrows represent the location of the primers that were used for PCR-based genotyping. (**b**) Verification of the CR-*SlJA2* mutant alleles using DNA sequencing analysis. We found deletions and insertions that spanned the two sgRNA target sites in the CR-*SlJA2*-5 and CR-*SlJA2*-20 lines. The red font indicates the sgRNA target sequences. The blue boxes indicate the protospacer-adjacent motif sequences. (**c**) The expression levels of *SlJA2* in 5 OE-*SlJA2*-lines. Three biological experiments from independent RNA extractions for each group of leaves were analyzed. (**d**) Root phenotypes and (**e**) root hair of CR-*SlJA2* and OE-*SlJA2* lines compared with WT (AC) lines subjected to low-K^+^ treatment with KNO_3_ (0.5 mM) as ‘LK’. The treatment time points were 0 and 7 d. Bars = 3 cm/100 μm. (**f**) The seedling aboveground phenotype of OE-*SlJA2* and CR-*SlJA2* lines after low-K^+^ treatment for 7 d. (**g**) The K^+^ content in the CR-*SlJA2* and OE-*SlJA2* lines determined at the CK and LK treatment time points 0 and 7 d. Asterisks indicate significant difference as determined using one-way ANOVA (**, *p* < 0.01).

**Figure 7 ijms-24-07058-f007:**
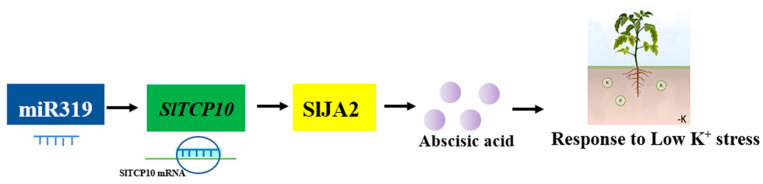
A simplified schematic model depicting how SlmiR319b and its targets, *SlTCP10* and *SlJA2*, integrate ABA signaling to regulate the response to low-K^+^ in tomato plants. Under low-K^+^ stress, miR319 targets *SlTCP10*, which directly interacts with *SlJA2*. Furthermore, *SlJA2* also induces endogenous ABA biosynthesis and ABA could influence the response to low-K^+^ stress.

**Table 1 ijms-24-07058-t001:** Comparison of the root parameter characteristics between JZ34 and miR319-OE transgenic tomato in low potassium. CK represents the control; LK represents the low-potassium treatment. Treatment time = 7 d.

		Genotype		LK Reduction Percentage	
Root Index	K^+^ Level	JZ34	miR319-OE	JZ34	miR319-OE
Length (cm)	CK	599.12 ± 33.21 a	463.16 ± 12.36 b	8.21%	12.04%
	LK	549.93 ± 43.29 a	407.40 ± 15.99 c		
Volume (cm^3^)	CK	1.22 ± 0.03 a	0.94 ± 0.11 b	18.85%	18.09%
	LK	0.99 ± 0.14 b	0.77 ± 0.06 c		
Surface area (cm^2^)	CK	109.47 ± 10.44 a	83.63 ± 9.49 b	11.99%	9.97%
	LK	96.34 ± 10.32 a	75.29 ± 8.03 c		

LK reduction percentage = (|LK – CK|/CK). Numbers are presented as mean ± SE. The number of observations in each mean is 3. Difference between WT and three transgenic lines, on average. Means in the same column followed by the same letter are not significantly different at *p* < 0.05.

**Table 2 ijms-24-07058-t002:** The number of the root hairs between JZ34 and OE-miR319b transgenic tomatoes in low potassium for 0 d, 3 d and 7 d. CK represents the control; LK represents the low-potassium treatment.

		Genotype		LK Reduction Percentage	
Root Index	K^+^ Level	JZ34	miR319-OE	JZ34	miR319-OE
Length (cm)	CK	599.12 ± 33.21 a	463.16 ± 12.36 b	8.21%	12.04%
	LK	549.93 ± 43.29 a	407.40 ± 15.99 c		
Volume (cm^3^)	CK	1.22 ± 0.03 a	0.94 ± 0.11 b	18.85%	18.09%
	LK	0.99 ± 0.14 b	0.77 ± 0.06 c		
Surface area (cm^2^)	CK	109.47 ± 10.44 a	83.63 ± 9.49 b	11.99%	9.97%
	LK	96.34 ± 10.32 a	75.29 ± 8.03 c		

LK reduction percentage = (|LK – CK|/CK). Numbers are presented as mean ± SE. The number of observations in each mean is 3. Difference between WT and three transgenic lines, on average. Means in the same column followed by the same letter are not significantly different at *p* < 0.05.

**Table 3 ijms-24-07058-t003:** The number of the root hairs of AC, OE-*SlJA2* and CR-*SlJA2* transgenic tomatoes in low potassium after 0 d and 7 d. LK represents the low-potassium treatment.

Root Index	K^+^ Level	Treat Time	Genotype				
			AC	CR-*SlJA2*#5	CR-*SlJA2*#20	OE-*SlJA2*#1	OE-*SlJA2*#2
Number of root hairs	LK	0 d	29.7 ± 8.2	10.0 ± 1.4	14.3 ± 0.9	120.7 ± 30.7	76.0 ± 12.2
		7 d	95.7 ± 8.7	4.3 ± 2.1	5.7 ± 3.3	130.7 ± 23.9	89.0 ± 12.8

Numbers are presented as mean ± SE. The number of observations in each mean is 3. Difference between WT and three transgenic lines, on average. Means in the same column followed by the same letter are not significantly different at *p* < 0.05.

**Table 4 ijms-24-07058-t004:** Comparison of the root parameter characteristics between AC, OE-SlJA2 and CR-SlJA2 transgenic tomatoes in low potassium. CK represents the control; LK represents the low-potassium treatment after 7 d.

Root Index	K^+^ Level	Genotype		
		AC	OE-*SlJA2*	CR-*SlJA2*
Length(cm)	CK	573.653 ± 13.836 a	636.085 ± 103.601 a	187.812 ± 7.981 b
	LK	548.232 ± 41.170 a	640.013 ± 6.053 a	145.622 ± 3.244 b
Volume(cm^3^)	CK	1.2575 ± 0.0125 a	2.574 ± 0.082 c	0.483 ± 0.017 e
	LK	1.582 ± 0.001 b	2.807 ± 0.042 d	0.441 ± 0.082 e
Surface area(cm^2^)	CK	95.209 ± 1.615 a	143.437 ± 13.495 b	33.284 ± 0.310 c
	LK	100.522 ± 0.145 a	149.039 ± 0.124 b	28.248 ± 2.320 c

Numbers are presented as mean ± SE. The number of observations in each mean is 3. Difference between WT and three transgenic lines, on average. Means in the same column followed by the same letter are not significantly different at *p* < 0.05.

## Data Availability

All data included in this study are available upon request by contact with the corresponding author.

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
