# Peer review of "Regulation of miR319b-Targeted SlTCP10 during the Tomato Response to Low-Potassium Stress"

_ijms, 2023, doi:10.3390/ijms24087058_

Round 1
Reviewer 1 Report
The manuscript (ijms-2276411) “Regulation of miR319b-targeted SlTCP10 during the tomato response to low potassium stress” showed that SlmiR319b and its target SlTCP10 mediate SlJA2 expression, thereby integrating 466 hormonal ABA signaling to regulate low-K+ tolerance in tomato plants. This study is a continuation of previous studies with sufficient data. The results will help us understand the response mechanism of miR319b-targeted SlTCP10 during the tomato response to low potassium stress.
Some minor problems need attention:
(1) The summary of suggestions should be reorganized. Phenotype-function-mechanism
(2) In the result section, please exchange the order of 2.2 and 2.3.
(3) Line103 to 106: Please combine the description.
(4) Line 107 two or three?
(5) The discussion section is a weak one, and seems to have only discussed SLJA2. There is too little discussion on miR319b and SlTCP10. It is necessary to compare them with the literatures and refine the novelty of your research.
(6) The literature needs to be supplemented with the latest relevant research progress. I checked that the latest year is 2018. In fact, there have been many registrations in recent years, such as https://doi.org/10.1016/j.cj.2020.07.007.
Author Response
Reviewer 1:
The manuscript (ijms-2276411) “Regulation of miR319b-targeted SlTCP10 during the tomato response to low potassium stress” showed that SlmiR319b and its target SlTCP10 mediate SlJA2 expression, thereby integrating 466 hormonal ABA signaling to regulate low-K+ tolerance in tomato plants. This study is a continuation of previous studies with sufficient data. The results will help us understand the response mechanism of miR319b-targeted SlTCP10 during the tomato response to low potassium stress.
Some minor problems need attention:
- The summary of suggestions should be reorganized. Phenotype-function-mechanism
Answer: We have followed your suggestions and reorganized the summary in the New Manuscript.
- In the result section, please exchange the order of 2.2 and 2.3.
Answer: We have followed your suggestions and exchange the order in the New Manuscript.
- Line103 to 106: Please combine the description.
Answer: We have followed your suggestions and combined the description in the New Manuscript.
(4) Line 107 two or three?
Answer: We have followed your suggestions and changed the ‘two’ to ‘three’ in the New Manuscript.
(5) The discussion section is a weak one, and seems to have only discussed SLJA2. There is too little discussion on miR319b and SlTCP10. It is necessary to compare them with the literatures and refine the novelty of your research.
Answer: We have followed your suggestions and the discussion on miR319-TCPs abiotic stress researches in the Line 325-331 of the New Manuscript.
(6) The literature needs to be supplemented with the latest relevant research progress. I checked that the latest year is 2018. In fact, there have been many registrations in recent years, such as https://doi.org/10.1016/j.cj.2020.07.007
Answer: We have followed your suggestions and added several latest relevant research in the New Manuscript.

Reviewer 2 Report
Please find my comments below: 1.) the authors aimed to understand the interaction of micro RNA 319 with SlTCP10, a transcription factor regulated under low potassium contents. miRNAs are regulators for gene expression, but the thier functions and interactions partners are for most miRNAs not yet described. Therefore the study is relevant to provide this kind of information for the specific mentioned microRNA. 2) Studying the regulation of a specific gene/transcription factor is difficult and not always straight forward. I, therefore, consider the presented results as relevant for the specific mentioned genes, as they provide information not yet known 3) Indeed the manuscript lacks a conclusive comparison with other miRNAs known to regulate/respond to low potassium conditions. The authors may add more recent work on this complex issue in the discussion and clarify also the aspect of the phytohormonal involvement in the regulation and response of the selected cultivars. 4) the cultivars JZ18 and JZ34 could be better described - as I understood, these are mutants deficient in ABA biosynthesis - the context related to the study should be clarified and why no wild type cultivar was used. Specify also the genetic transformation of tomatoes .- stable transformation or hairy roots - this is not clear from the method description 7) some figures could be of better quality, but maybe this is planned for the final editing. Use of a consistent number of decimal places in tables 5) maybe a concluding figure could help to summarize the results, as it is hard to follow all interactions and key results.Author Response
Reviewer 2:
Please find my comments below: 1.) the authors aimed to understand the interaction of microRNA 319 with SlTCP10, a transcription factor regulated under low potassium contents. miRNAs are regulators for gene expression, but the their functions and interactions partners are for most miRNAs not yet described. Therefore the study is relevant to provide this kind of information for the specific mentioned microRNA.
Answer: We have followed your suggestions and introduced the most miRNAs functions and interactions partners in the Introduction in the New Manuscript Line 38-43.
2) Studying the regulation of a specific gene/transcription factor is difficult and not always straight forward. I, therefore, consider the presented results as relevant for the specific mentioned genes, as they provide information not yet known
Answer: The targets of miR319 were mainly the TCP transcription factors. SlmiR319b overexpression lines actually changed the tomato tolerance to low-K+ stress. And it was found SlJA2 promoter has the TCP combined region. We also identified the interaction between SlTCP10 and SlJA2. Finally, the tolerance to low-K+ stress of CR-SlJA2 and OE-SlJA2 were defined. So this regulation pathway is workable and reliable.
3) Indeed the manuscript lacks a conclusive comparison with other miRNAs known to regulate/respond to low potassium conditions. The authors may add more recent work on this complex issue in the discussion and clarify also the aspect of the phytohormonal involvement in the regulation and response of the selected cultivars.
Answer: We have followed your suggestions and add more other miRNAs in Discussion of the New Manuscript.
4) the cultivars JZ18 and JZ34 could be better described - as I understood, these are mutants deficient in ABA biosynthesis - the context related to the study should be clarified and why no wild type cultivar was used. Specify also the genetic transformation of tomatoes .- stable transformation or hairy roots - this is not clear from the method description
Answer: Two tomato genotypes ‘JZ34’ (low K+-tolerant) and‘JZ18’ (low K+-sensitive) were obtained in our lab by higher generation inbred lines and introduced in detail about the low-K+ tolerance by Zhao et al. in 2018 and Liu et al. in 2020. And we redescribed these two genotypes tomatoes in the Materials and Methods of the New Manuscript Line 363-365.
5) some figures could be of better quality, but maybe this is planned for the final editing. Use of a consistent number of decimal places in tables
Answer: We have followed your suggestions and revised the decimal plants in Table 4 of the New Manuscript. And we uploaded the figures compression packages.
6) maybe a concluding figure could help to summarize the results, as it is hard to follow all interactions and key results.
Answer: We have followed your suggestions and added a concluding figure in Fig. 7 of the New Manuscript.
References:
Zhao XM, Liu Y, Liu X, Jiang J. Comparative transcriptome profiling of two tomato genotypes in response to potassium-deficiency stress. Int J Mol Sci. 2018,19: 2402.
Liu X, Tan C, Cheng X, Zhao X, Li T, Jiang J. miR168 targets Argonaute1A mediated miRNAs regulation pathways in response to potassium deficiency stress in tomato. BMC Plant Biology, 2020, 20: 477.
